

# Genome-wide analysis of bZIP transcription factors and their expression patterns in response to methyl jasmonate and low-temperature stresses in *Platycodon grandiflorus*

Jizhou Fan[1], Na Chen[2,3], Weiyi Rao[1,4], Wanyue Ding[1], Yuqing Wang[1], Yingying Duan[1], Jing Wu[1] and Shihai Xing[1,2,5]

[1] College of Pharmacy, Anhui University of Chinese Medicine, Hefei, Anhui, China
[2] Institute of Health and Medicine, Hefei Comprehensive National Science Center, Joint Research Center for Chinese Herbal Medicine of Anhui, Bozhou, Anhui, China
[3] College of Pharmacy, Bozhou Vocational and Technical College, Bozhou, Anhui, China
[4] Institute of Traditional Chinese Medicine Resources Protection and Development, Anhui Academy of Chinese Medicine, Hefei, Anhui, China
[5] Anhui Province Key Laboratory of Research and Development of Chinese Medicine, Anhui University of Chinese Medicine, Hefei, Anhui, China

Corresponding author
Shihai Xing, xshshihai@163.com

## ABSTRACT

**Background:** *Platycodon grandiflorus* belongs to the genus Platycodon and has many pharmacological effects, such as expectorant, antitussive, and anti-tumor properties. Among transcription factor families peculiar to eukaryotes, the basic leucine zipper (bZIP) family is one of the most important, which exists widely in plants and participates in many biological processes, such as plant growth, development, and stress responses. However, genomic analysis of the bZIP gene family and related stress response genes has not yet been reported in *P. grandiflorus*.

**Methods:** *P. grandiflorus bZIP* (PgbZIP) genes were first identified here, and the phylogenetic relationships and conserved motifs in the PgbZIPs were also performed. Meanwhile, gene structures, conserved domains, and the possible protein subcellular localizations of these PgbZIPs were characterized. Most importantly, the cis-regulatory elements and expression patterns of selected genes exposed to two different stresses were analyzed to provide further information on PgbZIPs potential biological roles in *P. grandiflorus* upon exposure to environmental stresses.

**Conclusions:** Forty-six PgbZIPs were identified in *P. grandiflorus* and divided into nine groups, as displayed in the phylogenetic tree. The results of the chromosomal location and the collinearity analysis showed that forty-six PgbZIP genes were distributed on eight chromosomes, with one tandem duplication event and eleven segmental duplication events identified. Most PgbZIPs in the same phylogenetic group have similar conserved motifs, domains, and gene structures. There are cis-regulatory elements related to the methyl jasmonate (MeJA) response, low-temperature response, abscisic acid response, auxin response, and gibberellin response. Ten PgbZIP genes were selected to study their expression patterns upon exposure to low-temperature and MeJA treatments, and all ten genes responded to these stresses. The real-time quantitative polymerase chain reaction (RT-qPCR) results suggest that the expression levels of most PgbZIPs decreased significantly

within 6 h and then gradually increased to normal or above normal levels over the 90 h following MeJA treatment. The expression levels of all PgbZIPs were significantly reduced after 3 h of the low-temperature treatment. These results reveal the characteristics of the PgbZIP family genes and provide valuable information for improving *P. grandiflorus's* ability to cope with environmental stresses during growth and development.

## INTRODUCTION

*Platycodon grandiflorus* (Jacq.) A. DC. is the sole member of the Campanulaceae family, mainly found in northeastern, northern, and central China, South Korea, Japan, the Russian Far East, and southeastern Siberia (*Liu et al., 2021*; *Wang et al., 2017b*; *Zhang et al., 2022*). *P. grandiflorus* roots are widely used in treatments such as analgesia, digestive tract ulcers, cough, and hyperglycemia because of their significant efficacy in immunostimulatory, anti-inflammatory, antioxidant, and anti-tumor activities (*Choi et al., 2010*; *Lee et al., 2020*; *Wang et al., 2017a*; *Zhao et al., 2018*). It has high medicinal value, and demand for it continues to increase, but there is a lack of wild germplasm resources, so scientific artificial cultivation and planting are overwhelmingly important. Artificial cultivation has been commonly used because it lacks wide germplasm resources. Nevertheless, environmental stress is of great significance to plant growth and development.

The basic leucine zipper (bZIP) supergene family encodes transcription factors and is widely distributed in eukaryotes. The transcription factor proteins encoded by the bZIP gene family contain the highly conserved bZIP domain. This domain is characterized by a basic DNA binding domain and an adjacent leucine zipper, allowing bZIP to dimerize (*Kouzarides & Ziff, 1989*; *Vinson, Sigler & McKnight, 1989*). It contains 60–80 amino acids and consists of two regions with different functions: the alkaline region and the leucine zipper region (*Correa et al., 2008*; *Jakoby et al., 2002*; *Nijhawan et al., 2008*). The alkaline region comprises approximately 16 amino acid residues, which are highly conserved at the N-terminal of the bZIP DNA binding domain (*Correa et al., 2008*; *Foster, Izawa & Chua, 1994*). The relatively less conserved leucine zipper region consists of seven repetitive sequences of leucine or other large hydrophobic amino acid residues located at the C-terminal of bZIP (*Droge-Laser et al., 2018*; *Jakoby et al., 2002*) and participates in bZIP protein dimerization before DNA binding. The bZIP protein usually plays a role through the leucine zipper to form dimers (*Liu et al., 2023*; *Sornaraj et al., 2016*).

bZIP transcription factors are widely found in eukaryotes such as plants, mammals, and insects but differ in number. At present, they are found in humans (53 members) (*Rodriguez-Martinez, Sossa-Briceno & Castro-Rodriguez, 2018*), rice (92 members) (*Correa et al., 2008*), corn (125 members) (*Wei et al., 2012*), and *Arabidopsis thaliana* (78 members) (*Droge-Laser et al., 2018*). Recently, some researchers have updated the classification of *A. thaliana* bZIPs (AtbZIPs) and divided them into 13 groups (*Droge-*

*Laser et al., 2018*). Some studies have shown that segmental genome duplication and whole genome duplication (WGD) events may be the reasons for the expansion of the bZIP gene family (*Nijhawan et al., 2008*; *Zhou et al., 2017*). At the same time, bZIP gene functions in plants have attracted the attention of many researchers. This gene family plays an important role in many biological processes, such as plant growth, development, and stress responses (*Noman et al., 2017*; *Shearer et al., 2012*; *Zhao et al., 2016*). The over-expression of *A. thaliana FD* and *FDP* in rice led to reduced plant height and minor panicles in transgenic rice (*Jang, Li & Kuo, 2017*). Moreover, the over-expression of *ZmbZIP4* in maize has been shown to increase the number of lateral roots (*Ma et al., 2018*). In soybeans, three bZIP genes were found to be negative regulators of abscisic acid signaling and were endowed with salt and cold tolerance in transgenic *A. thaliana* (*Liao et al., 2008*). However, there is no research on the bZIP gene family in *P. grandiflorus*.

Methyl jasmonate (MeJA) is an important derivative of jasmonic acid, which plays a crucial regulatory role in different plant development and signal network transmission processes (*Rahnamaie-Tajadod, Goh & Mohd Noor, 2019*). On the one hand, MeJA can help plants resist biotic and abiotic stresses and affect physiological and biochemical functions (*Yu et al., 2019*). On the other hand, it also stimulates molecular signal transduction and regulates gene expression, leading to secondary metabolite accumulation (*Rahimi, Kim & Yang, 2015*). Since MeJA regulates plant physiological and biochemical processes and affects primary and secondary metabolite synthesis, it has been widely used in improving horticulture, crop yield, fruit quality and taste, and bioactive components in medicinal plants (*Luo et al., 2020*; *Ramabulana et al., 2020*). Low-temperature stress seriously affects plant growth and development. In addition, chilling injuries can lead to difficult seed germination, weak seedlings, plant growth and development retardation, wilting, yellowing, low seed setting rate, and lower yield and quality, which does great harm to plant growth and development. Many plant transcription factors respond to low-temperature stress, and the expression levels of these transcription factors is, to some extent, regulated by low temperatures (*Yao et al., 2022*; *Zhang et al., 2023*). Previous studies have shown that bZIP transcription factors are also involved in hormone and light signal transmission, including abscisic acid and gibberellin signaling, and can effectively induce resistance to severe cold, drought, saline–alkali, and other harsh environments and enhance plant resistance to stress by regulating related gene expression. The optimum cultivation conditions for *P. grandiflorus* are between 10 °C and 35 °C, with growth inhibition occurring below 20 °C. The platycodins, which are the main active components in *P. grandiflorus*, may be induced by MeJA. However, whether MeJA and low-temperature stress regulate *P. grandiflorus bZIPs* (PgbZIP) gene expression pattern needs to be studied further (*Zhao et al., 2021*; *Zhou & Yarra, 2022*).

In this study, the whole *P. grandiflorus* genome was analyzed, and forty-six PgbZIP genes were identified. At the same time, we conducted genome-wide identification and comprehensive evolutionary analysis of PgbZIP transcription factors and predicted the functions of various PgbZIP family protein subclasses. In addition, to study the PgbZIP gene responses to environmental stress, real-time quantitative polymerase chain reaction (RT-qPCR) was used to quantify and compare the PgbZIP gene expression levels upon
exposure to low-temperature and MeJA stress treatments. The results of this study can provide a reference for improving *P. grandiflorus* low-temperature tolerance, and PgbZIPs response to MeJA was also investigated, which can provide a theoretical basis for the functional validation of PgbZIPs in the future.

## MATERIALS AND METHODS

### Plants and materials

The experimental materials were *P. grandiflorus* tissue culture seedlings grown in a growth chamber in 1/2 MS culture medium for 3 months (16 h/8 h light/dark cycle, 25 °C). The *P. grandiflorus* specimens were kept in the Plant Herbarium at Anhui University of Chinese Medicine (Hefei, Anhui, China). The *P. grandiflorus* seeds came from the School of Pharmacy, Anhui University of Chinese Medicine, specimen number 2020070 (*Su et al., 2021*). Healthy seedlings were randomly selected and divided into stress treatment groups in a climate chamber. In the MeJA treatment, MeJA (0.5 mM) was sprayed on *P. grandiflorus* seedlings with Tween 20 as a dispersant. *P. grandiflorus* samples were collected at 0, 6, 24, and 96 h after MeJA treatment. In the low-temperature treatment, the plants were stored in an incubator at 4 °C, and the *P. grandiflorus* samples were collected at 0, 1, 3, 6, 9, and 12 h after treatment. Each sample in this study had three independent biological duplicates, and the collected samples were immediately stored at −80 °C and used in the experiment within 12 h.

### Identification of bZIPs in *P. grandiflorus*

The genome database, the protein database, and related annotation files for *P. grandiflorus* were downloaded from the National Genomics Data Center (https://ngdc.cncb.ac.cn), ID: PRJCA003843 (*Jia et al., 2022*), and the hidden Markov model (HMM) files for bZIP transcription factor conserved domains (PF07716, PF00170, PF03131, and PF16326) were downloaded from the Pfam database (http://pfam.xfam.org/) (*Li et al., 2021*). AtbZIP family transcription factors were downloaded from the Plant Transcription Factor Database (http://planttfdb.gao-lab.org/). In the BioEdit software, the *P. grandiflorus* protein database was used to blast with AtbZIP transcription factors and HMM models, and the E-value was set to 0.000001. The obtained sequences were merged, and the redundant sequences were removed using CD-HIT online analysis (http://cd-hit.org/) (*Li & Godzik, 2006*). Candidate PgbZIP proteins were subjected to multiple sequence alignments in the MEGAX software to find conserved residues specific to the bZIP family, and proteins that did not possess conserved residues were deleted. Then, the physical and chemical properties of these protein sequences were evaluated using the ExPASy5 ProtParam tool (*Savojardo et al., 2018*), and the protein subcellular localizations were predicted using the Busca online website (http://busca.biocomp.unibo.it/) (*Gasteiger et al., 2003*).

### Constructing phylogenetic trees of bZIPs in *P. grandiflorus*

To analyze the evolutionary relationships of conserved protein sequences, we constructed a phylogenetic tree using the MEGAX software for bZIP transcription factor protein

sequences obtained from *P. grandiflorus* and *A. thaliana* (*Kumar et al., 2018*). Sequences were aligned using Clustal W method, and evolutionary trees were generated using the neighbor-joining method (Bootstrap values set to 1,000, pairwise deletion of gaps, and the p-distance method were used) in MEGAX.

## Chromosomal localization and gene duplication events of PgbZIPs

The inferred position information and chromosome lengths of bZIP genes were obtained from the general feature format file of the *P. grandiflorus* genome. According to the related literature (*Wang et al., 2012*), the chromosome location and the collinearity analysis were carried out using MG2C (http://mg2c.iask.in/mg2c_v2.1/) and the TB-tool software. We use the Quick Genome gene dot plot tool to describe gene duplication events and the Simple Ka/Ks Calculator tool for the Ka/Ks analysis.

## Identification of the PgbZIP conserved motifs

The conserved PgbZIP motifs were analyzed using the online website MEME (http://meme-suite.org/tools/meme) (*Bailey et al., 2015*). The predicted value was set to 20, and the conserved PgbZIP motifs were obtained. The parameters are set as follows: motif sites are distributed sequentially, and each sequence has a contributing motif site; the maximum number that needs to be found is 20; the minimum and maximum motif widths are between 10 and 50; the motif value is retained $<1 \times 10^{-20}$; the default value was used for other options for further analysis.

## Identification of cis-regulatory elements in PgbZIP gene promoters

The bZIP gene regulatory regions at 2 kb upstream were extracted from the *P. grandiflorus* coding sequence using the TB-tool software (*Chen et al., 2020*). The promoter sequence's cis-regulatory elements were detected using the TBtools software, and the same software was used to display these elements.

## Real-time quantitative PCR

Using the TransZol kit (TransGen Biotech, Inc., Beijing, China), the total RNA of *P. grandiflorus* was extracted according to the manufacturer's instructions. cDNA was synthesized using a FastKing RT Kit (Tiangen, Beijing, China) and reverse transcription reaction. The primers were designed using the Primer 5.0 software (Table S1). Gel electrophoresis showed the RNA integrity, and the RNA concentration was detected using a Nanodrop microspectrophotometer. The RNA purity was good, RIN > 7, and the A260/280 ratio was between 1.9 and 2.1. Optimus Biotechnology Co., Ltd. synthesized the primers, and the 18s rRNA gene was used as the internal reference gene (*Su et al., 2021*). The RT-qPCR analysis was performed using a SuperReal PreMix Plus SyBr Green PCR kit (Qiagen, Hilden, Germany) on a Cobas z480 Real-Time PCR System using the method described by *Su et al. (2021)*. The thermal program of the PCR is as follows: pre-incubation at 95 °C for 15 min; three steps for amplification over 50 cycles, 95 °C for 10 s, 58 °C for 20 s, and 72 °C for 20 s. Each reaction was repeated three times, and a negative control group was set up at 0 h. The relative gene expression was calculated using the $2^{-\Delta\Delta Ct}$ method. One-way analysis of variance (ANOVA) was applied for data processing, and statistical

differences were compared using t-tests based on IBM SPSS Statistics 23. Finally, visual analysis was performed using GraphPad Prism 8.0.1 (GraphPad Software, La Jolla, CA, USA).

## RESULTS

### Analysis of PgbZIP transcription factors

According to the similarity to AtbZIPs and the HMM of the bZIP–DNA binding domain, a total of 75 proteins were found in *P. grandiflorus*. The bZIP basic region contains ~16 amino acid residues. The most striking characteristic of this basic region is the presence of invariant N-x7-R/K conserved residues (*Lai et al., 2024*). Through multiple sequence alignments of the 75 proteins, we found that only 46 proteins contain the typical N-x7-R/K conserved residues (Fig. S1). These proteins are named *PgbZIP001* to *PgZIP046* (Table S2). According to the analysis of protein sequence characteristics, the lengths of these 46 PgbZIP proteins varied between 139 and 730 amino acids; the relative molecular weights were between 16.46 and 78.77 kDa; the average relative molecular weight was 36.38 kDa; the isoelectric points were between 4.65 and 10.23; the average isoelectric point was 7.17. All PgbZIPs have negative grand average of hydropathy (GRAVY) values, which indicates that PgbZIP proteins are hydrophilic proteins. The subcellular localization prediction showed that PgbZIPs are located in the nucleus (45) and cytoplasm (one) (Table S3).

### Phylogenetic evolution and classification of PgbZIPs

*A.thaliana* is one of the earliest plants studied for the bZIP transcription factor family, holds significant reference value, and is commonly employed as a model genetic plant for plant genetic research (*Droge-Laser et al., 2018*; *Jakoby et al., 2002*). To analyze the phylogenetic relationship of these PgbZIPs, we constructed a phylogenetic tree using 127 AtbZIP proteins and 46 PgbZIP proteins (Fig. 1). According to the reports of *A. thaliana*, the PgbZIP phylogenetic tree was divided into nine groups (A-E, G-I, and S) (*Droge-Laser et al., 2018*). In the PgbZIP family, the largest groups are Groups S and A (each with ten members), and the smallest group is Group H (one member).

### Chromosomal location and collinearity analysis of PgbZIPs

These PgbZIP genes were physically located on the chromosome in *P. grandiflorus* (Pgchr) using the MG2C online tool (Fig. 2A). The PgbZIP lengths and positions on the chromosomes are shown in Table S3. A total of nine chromosomes were shown in *P. grandiflorus* genomic data. The results showed that these forty-six PgbZIPs were distributed on eight Pgchrs. Among them, most PgbZIPs were found on Pgchr1, Pgchr2, and Pgchr3 (each with eight members), and the smallest number of PgbZIPs (two members) were found on Pgchr9. This PgbZIP gene distribution pattern on chromosomes may be related to differences in chromosome structure and size.

   Gene duplication events are important in promoting plant evolution, and new gene functions are derived through this process. In this study, we analyzed gene duplication events during *P. grandiflorus* evolution (Figs. 2B and 2C). Gene duplication event analysis uncovered 1,476 tandem duplication events in *P. grandiflorus*, but among them, only one

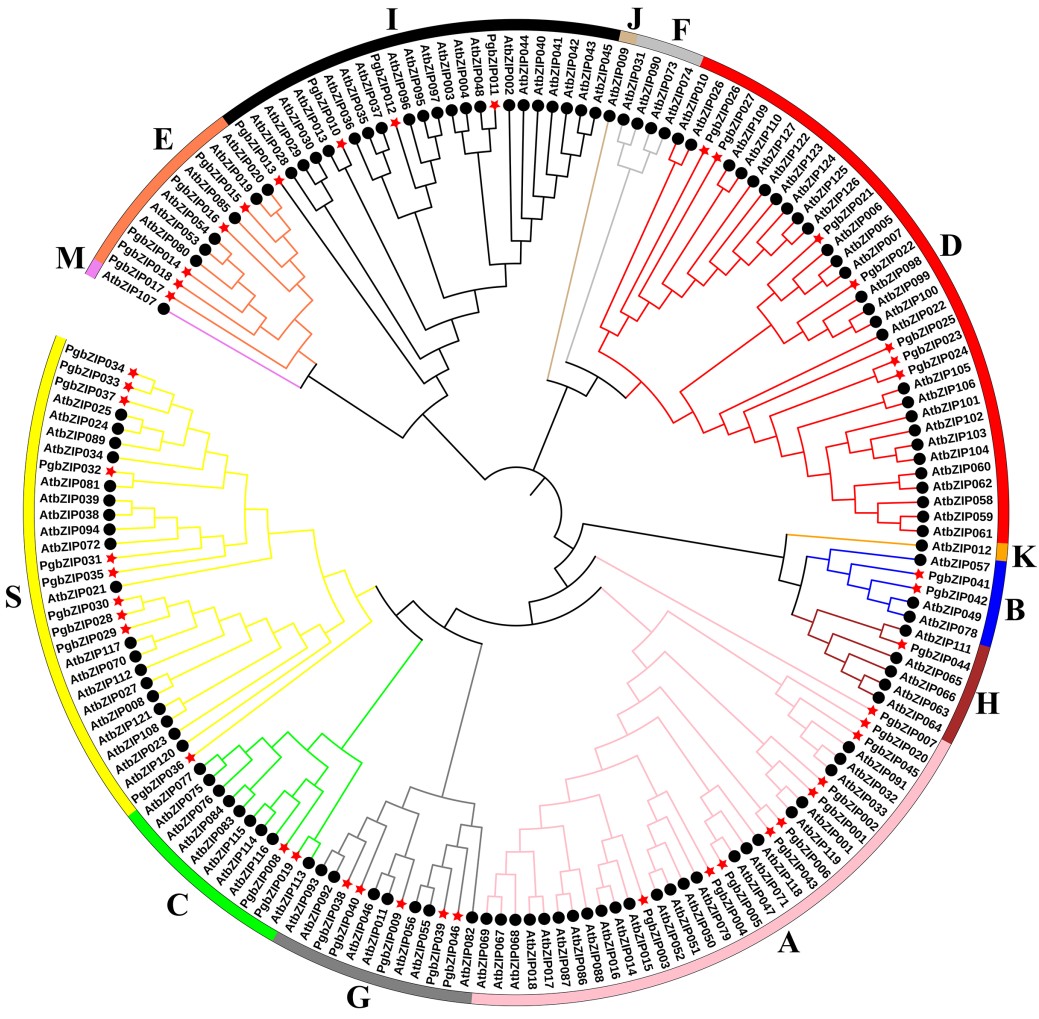

**Figure 1 Phylogenetic tree of bZIPs in *P. grandiflorus* and *A. thaliana*.** Different colors were assigned to distinguish different groups. Red asterisks represent *P. grandiflorus*, and black circles represent *A. thaliana*.

tandem duplication event was identified on Pgchr2 among the PgbZIPs studied (Fig. 2C). Collinearity analysis showed 1,913 suspected gene duplication events during *P. grandiflorus* evolution. After further analysis, 1,689 duplication events were found to be WGDs or segmental duplications. The dot map shows that most WGD events occurred on Pgchr1, Pgchr2, Pgchr5, Pgchr6, and Pgchr7 (Fig. 2B). In addition, 11 gene segmental duplication events were identified between PgbZIPs (Fig. 2C). The frequency of PgbZIP gene segmental duplication events was the highest on Pgchr1, with seven gene segmental duplication events, followed by Pgchr2 with five gene segmental duplication events. As shown in the Fig. 2C, the PgbZIP genes between Pgchr1 and Pgchr2 segmental were duplicated most frequently and have the strongest homology. For a gene that is not under pressure from natural selection, the value of Ka/Ks is close to 1. The Ka/Ks analysis showed that the Ka/Ks of segmental duplication gene pairs was much less than 1 (Table 1). Therefore, we can judge that they have a strong purifying selection.

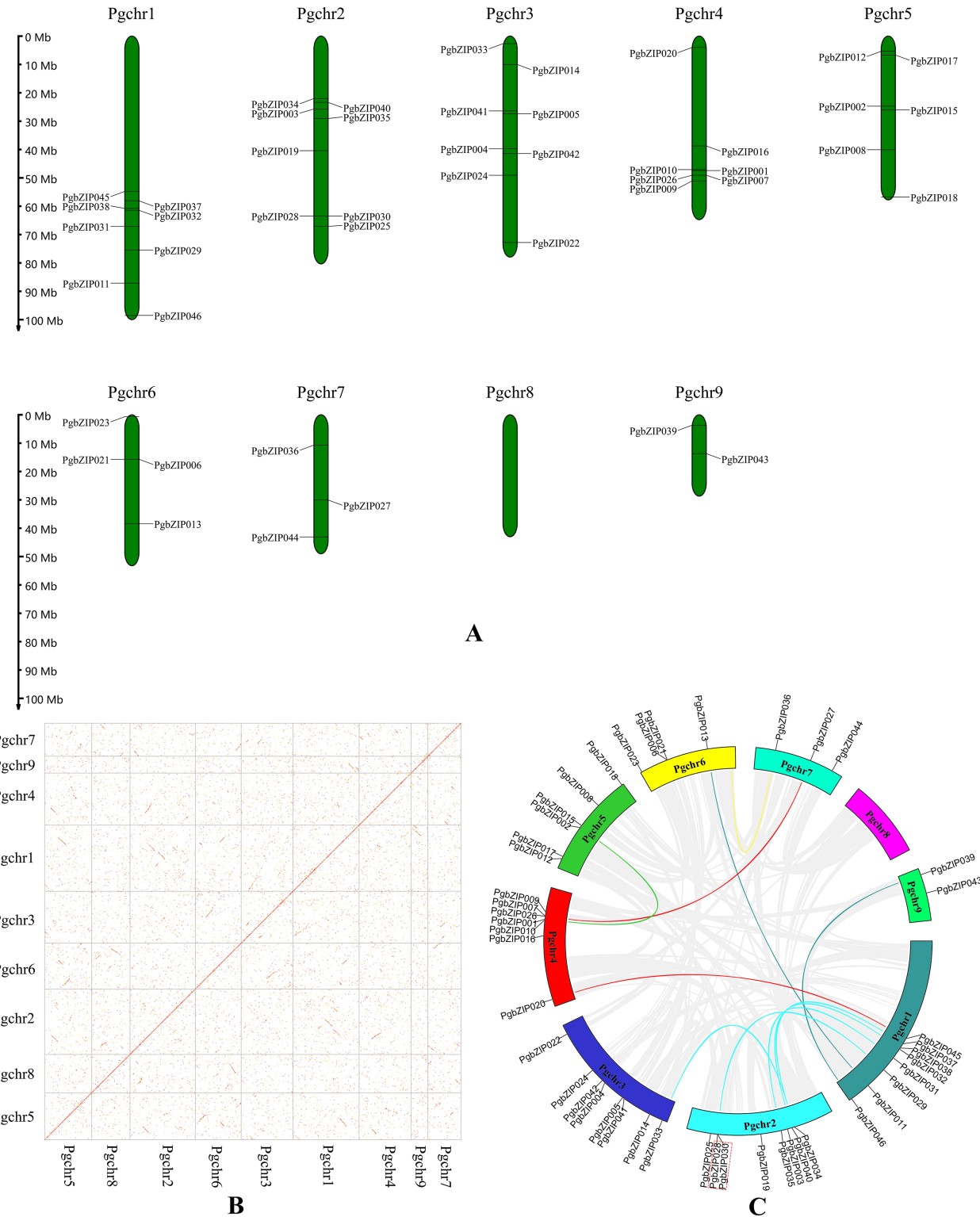

**Figure 2 Chromosomal location and collinearity analysis of *PgbZIPs*.** (A) Chromosomal location of PgbZIPs. (B) The genome dotplot of *P. grandiflorus*. The dotted lines denote WGD duplication events that occur between two chromosomes. (C) Collinearity analysis of PgbZIPs. Colored lines in the middle indicate duplication of PgbZIP gene pairs, whereas gray lines indicate genome duplication of gene pairs. The red dashed box indicates tandem duplication.

| Table 1 | Ka/Ks ratio among PgbZIPs. | | | |
|---|---|---|---|---|
| Seq 1 | Seq 2 | Ka | Ks | Ka/Ks |
| PgbZIP020 | PgbZIP045 | 0.5837 | 2.8119 | 0.2076 |
| PgbZIP034 | PgbZIP037 | 0.3228 | 1.9126 | 0.1688 |
| PgbZIP034 | PgbZIP033 | 0.3030 | 1.4205 | 0.2133 |
| PgbZIP040 | PgbZIP038 | 0.4476 | 1.5008 | 0.2982 |
| PgbZIP035 | PgbZIP031 | 0.4431 | 1.9001 | 0.2384 |
| PgbZIP030 | PgbZIP029 | 0.3272 | 1.6382 | 0.1997 |
| PgbZIP002 | PgbZIP001 | 0.2772 | 1.6312 | 0.1699 |
| PgbZIP026 | PgbZIP027 | 0.2720 | 1.1831 | 0.2299 |
| PgbZIP011 | PgbZIP013 | 0.4630 | 1.4789 | 0.3130 |
| PgbZIP046 | PgbZIP039 | 0.4954 | 1.0271 | 0.4823 |
| PGRA_17518 | PgbZIP036 | 0.6116 | 2.1259 | 0.2877 |

## Conserved motifs, conserved domains, and gene structures of PgbZIPs

The existence of motifs in the same phylogenetic group of PgbZIPs was similar (Figs. 3A and 3B). Twenty motifs were identified in PgbZIPs, ranging from six to 50 amino acids in length (Figs. 3B and 3E). Motif 2, Motif 3, Motif 4, Motif 5, Motif 10, Motif 11, and Motif 16 were found mainly in PgbZIPs from Group D. Motif 6, Motif 15, and Motif 20 were found mainly in PgbZIPs from Groups E and I. Motif 13, Motif 14, Motif 17, and Motif 19 were found mainly in PgbZIPs from Group G. Motif 8, Motif 9, and Motif 12 are found mainly in PgbZIPs from Group A, and Motif 18 is found mainly in PgbZIPs from Group S. Motif 7 is shared by Groups B, H, G, A, C, and S. Moreover, Motif 1 is widely found in all PgbZIP groups. Motif 1 included N-x7-R/K conserved residues so that we can find them in *PgbZIP001–PgbZIP045*. Motif 1 is not present in *PgbZIP046*, but in the conserved domain analysis, we found that *PgbZIP046* has the multifunctional mosaic region (MFMR) superfamily domain. The MFMR region contains a nuclear localization signal in bZIP opaque, and this region is found in the N-terminus of the bZIP transcription factor domain.

Conserved domain analysis reveals that the existence of domains in the same phylogenetic group of PgbZIPs was similar as well (Fig. 3C). The bZIP_HBP1b-like domain, shared by most PgbZIPs in Group D, contains a C-terminal DOG1 domain that is a specific seed dormancy plant regulator (*Mikami, Sakamoto & Iwabuchi, 1994*). The bZIP_plant_RF2 domains are owned by the PgbZIPs in both Groups E and I. Transcription factors with bZIP_plant_RF2 domains are similar to *Oryza sativa* RF2a and RF2b, and transgenic rice with increased RF2a and RF2b display increased resistance to rice tungro disease with no impact on plant development (*Dai et al., 2004*). The bZIP_plant_BZIP46 domain, present in most PgbZIPs in Group A, is similar to *GmBZIP46*, which may be a drought-response gene (*Liao et al., 2008*). The MFMR domain is possessed by the PgbZIPs in Group G, and it has been suggested that some of these

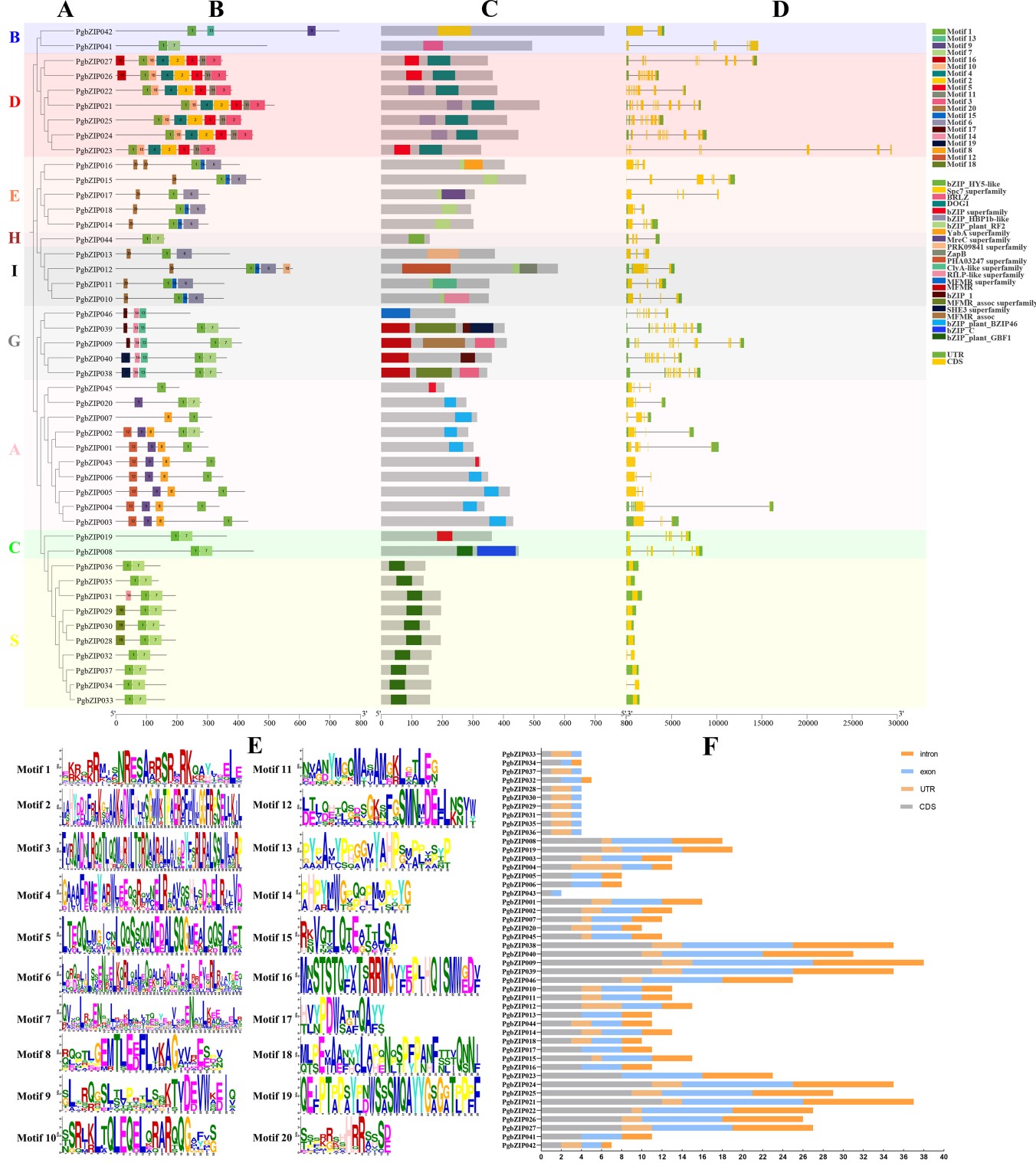

**Figure 3 Conserved motifs, conserved domains, and gene structures of PgbZIPs.** (A) Phylogenetic tree of PgbZIPs. Different colors were assigned to distinguish different groups. (B) Motifs in the PgbZIPs. Different colored squares represent different motifs, and the numbers on them represent the motif number. (C) Domains in the PgbZIPs. Different colored squares represent different domains. (D) Gene structures of PgbZIPs. Exon/intron structures of PgbZIPs are displayed by yellow bars and black lines, respectively. (E) Motif 1-Motif 20. (F) Statistical analysis of intron, exon, coding sequence (CDS), and untranslated regions (UTR) in PgbZIPs.

domains may be involved in mediating protein–protein interactions (*Sibéril, Doireau & Gantet, 2001*). The bZIP_plant_GBF1 domain of PgbZIPs in Group S is a DNA-binding and dimerization domain (*Norén Lindbäck et al., 2023*). The bZIP_plant_GBF1 domain contains an N-terminal proline-rich domain in addition to the bZIP domain. The transcription factors with this domain are involved in developmental and physiological processes in response to stimuli such as light or hormones and can also affect lysine content and carbohydrate metabolism, acting indirectly on the starch/amino acid ratio.

To reveal the PgbZIP gene family's structural evolution, the PgbZIP exon–intron structures were analyzed (Figs. 3D and 3F). The PgbZIP gene structures displayed significant variability and diversity in terms of the relative positions and numbers of introns and exons. PgbZIPs in Group B contained 1–3 introns, PgbZIPs in Group D contained 7–11 introns, PgbZIPs in Groups A and E contained 2–4 introns, PgbZIPs in both Groups H and I contained three introns, PgbZIPs in Group G contained 7–11 introns, PgbZIPs in Group A contained 0–4 introns, PgbZIPs in Group C contained five introns, and PgbZIPs in Group S contained 0–1 introns. PgbZIPs with close phylogenetic relationships may possess similar gene structures.

The similar domains, motifs, and gene structures in the same PgbZIP subgroups may lead to their similar functions, and the above results may provide some theoretical basis for future PgbZIP functional studies.

## Cis-regulatory elements of PgbZIPs

To explore PgbZIP gene responses to various stresses, a 2,000 bp DNA sequence region upstream of PgbZIPs was retrieved as the promoter region and used to analyze the PgbZIPs' cis-regulatory elements (Fig. 4). The results showed that among the PgbZIP gene promoters, the number of light responsive elements was the largest (518), followed by MeJA responsive elements (126). In addition, the numbers of abscisic acid responsive elements (90), anaerobic induction element (88), and MYB binding sites (57) are also extremely high. Notably, the low-temperature responsive elements (33) were the second most numerous in addition to the light responsive elements for natural environmental stresses.

## Expression pattern of *PgZIP* genes upon exposure to low-temperature and MeJA stresses

There is usually a correlation between gene expression pattern and function, and plant bZIP transcription factors are widely involved in various stresses. To investigate the response patterns of PgbZIPs to MeJA and low temperatures, *PgbZIP003* (eight), *PgbZIP005* (six), *PgbZIP02* (four), *PgbZIP021* (two), and *PgbZIP011* (zero) with different numbers of MeJA responsive elements and *PgbZIP046* (five), *PgbZIP019* (three), *PgbZIP041* (two), *PgbZIP021* (one), and *PgbZIP032* (zero) with different numbers of low-temperature responsive elements were selected to study their gene expression patterns upon exposure to low-temperature and MeJA stresses. RT-qPCR was conducted at 0, 6, 24, and 96 h after MeJA treatment and 0, 1, 3, 6, 9, and 12 h after low-temperature treatment (Fig. 5). The expression levels of nine PgbZIPs (*PgbZIP003, PgbZIP005, PgbZIP011,*

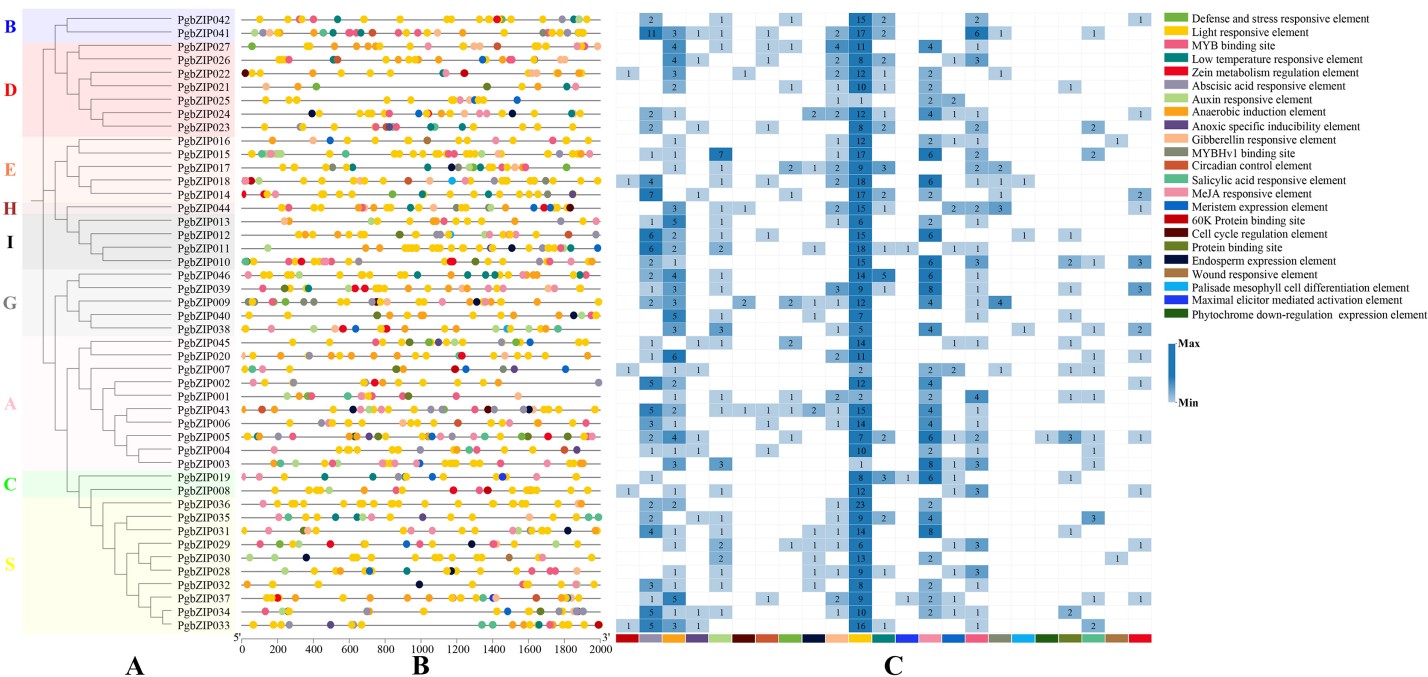

**Figure 4 Cis-regulatory elements of PgbZIPs.** (A) Phylogenetic tree of PgbZIPs. Different colors were assigned to distinguish the different groups. (B) Cis-regulatory elements of PgbZIPs. Different colored blocks represent different cis-regulatory elements. (C) Heatmap analysis of PgbZIP cis-regulatory elements. Numbers represent the number of cis-regulatory elements.

*PgbZIP019*, *PgbZIP021*, *PgbZIP022*, *PgbZIP024*, *PgbZIP032*, *PgbZIP042*, and *PgbZIP046*) were significantly decreased up to 6 h of MeJA treatment and gradually increased again after 6 h. The expression levels of seven PgbZIPs (*PgbZIP003*, *PgbZIP005*, *PgbZIP019*, *PgbZIP021*, *PgbZIP024*, *PgbZIP042*, and *PgbZIP046*) remained significantly lower than that of untreated (0 h) when treated with MeJA for 24 h. However, when the treatment time reached 96 h, most PgbZIPs (*PgbZIP011*, *PgbZIP019*, *PgbZIP021*, *PgbZIP024*, *PgbZIP042*, and *PgbZIP046*) returned to the untreated expression level or higher than that of the untreated expression level. In addition, *PgbZIP032* expression increased gradually after MeJA treatment. The expression levels of most PgbZIPs may decrease significantly (up to four-fold) within 6 h and then gradually increase to normal or above normal levels within 90 h after MeJA treatment. This result indicated that MeJA may have an inhibitory effect on the PgbZIP gene in the early response stage but increased PgbZIP gene expression after some time.

After low-temperature treatment, the expression trends of PgbZIPs was generally decreasing from 0 to 12 h. *PgbZIP003*, *PgbZIP005*, *PgbZIP011*, and *PgbZIP032* were all significantly lower than untreated at 12 h after the low-temperature treatment. At the same time, the expression levels of the other six PgbZIPs were also significantly lower than untreated at most time points after treatment. Notably, the expression levels of all PgbZIPs were significantly reduced after 3 h of the low-temperature treatment, with the largest decrease of about 35-fold for *PgbZIP032*. This shows that PgbZIP gene expression levels may be inhibited by low-temperature stress, and the occasional increase in expression level may be caused by *P. grandiflorus* adaptation to the low-temperature environment.

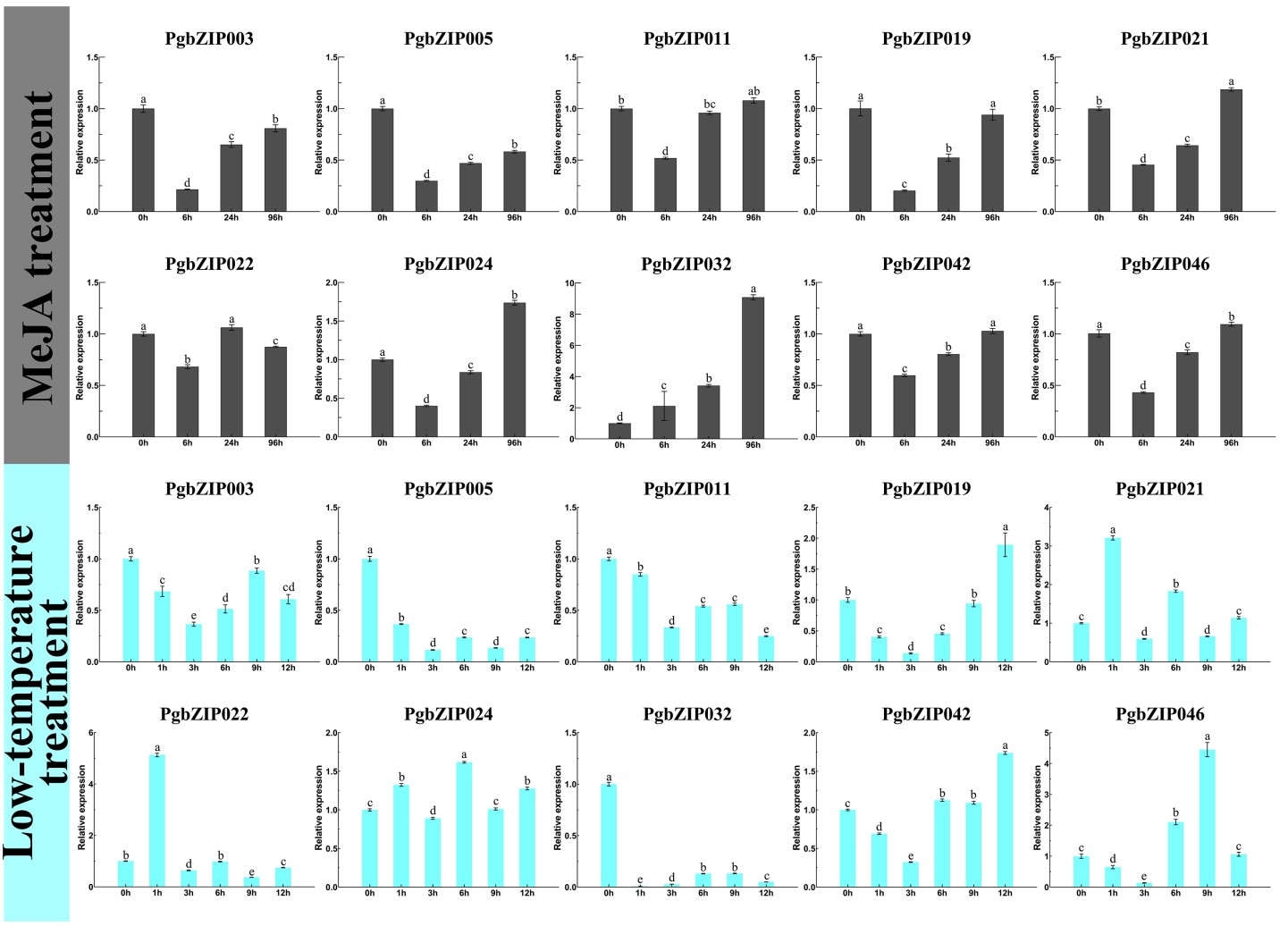

**Figure 5 Expression levels of PgbZIPs gene upon exposure to the low-temperature and MeJA treatments.** Error bars indicate the standard deviation (SD) of three biological replicates. Letters delineate significant differences among treatments ($p < 0.05$).

## DISCUSSION

The bZIP gene family is highly conserved in *P. grandiflorus*, and its expansion may be mainly dependent on segmental duplication.

According to the phylogenetic tree analysis with *A. thaliana*, PgbZIPs are divided into nine groups (A-E, G-I, and S). In previous studies, *Droge-Laser et al. (2018)* classified 127 AtbZIPs into 13 groups (A-K, M, and S). *Yue et al. (2023)* classified 161 *Glycine max* bZIPs (GmbZIPs) into 13 groups by performing a phylogenetic tree analysis of 161 GmbZIPs and 127 AtbZIPs ( A-K, M, and S). *Wang et al. (2023)* performed a phylogenetic tree analysis of 68 *Litsea cubeba* bZIPs (LcbZIPs) and 127 AtbZIPs and classified 68 LcbZIPs into 12 groups (A-K and S). *Lai et al. (2024)* performed a phylogenetic tree analysis of 70 *Cymbidium ensifolium* bZIPs (CebZIPs) and 78 AtbZIPs and classified 70 CebZIPs into 11 groups (A-I, K, and S). There are fewer PgbZIP subgroups compared with other species,

which may indicate that PgbZIPs are more conserved during evolution. Predictions of PgbZIPs' subcellular localization show that most PgbZIPs are located in the nucleus (97.83%), similar to other studies about plants (*Jakoby et al., 2002*; *Kouzarides & Ziff, 1989*; *Vinson, Sigler & McKnight, 1989*). All PgbZIPs had N-x7-R/K conserved residues, and further analysis revealed that different PgbZIP subgroups had similar conserved domains, motifs, and gene structures. The effects of different subgroups' specific domains, motifs, and gene structures on function are significant (*Yue et al., 2023*). The high degree of similarity in the structural features of these genes may be attributable to gene duplication events during the gene family expansion (*Yáñez et al., 2009*). Eleven fragment duplication events were identified in the PgbZIP genes, and these genes all have similar gene structures. Differences in gene structure arise from mutations in introns, such as base substitutions, insertions, and deletions, which alter the gene sequence and result in diversity in the number and arrangement of introns, contributing to the functional development of PgbZIP proteins (*Wang, Tan & Paterson, 2013*). The expansion of the transcription factor family in plants is likely caused by chromosomal, segmental, and tandem duplication, as well as transposition and homing (*Nakano et al., 2006*). A pair of tandem duplication genes and 11 pairs of fragment duplication genes were identified in PgbZIPs. This suggests that fragment duplication events may be one of the main reasons for PgbZIP gene family expansion.

## PgbZIPs response to low-temperature and MeJA stresses

The bZIP genes are involved in a variety of abiotic stresses in plants, including low-temperature, drought, and high-temperature stresses, and biotic stresses, such as diseases and pathogens, as well as in a variety of hormone-induced processes (*Amorim et al., 2017*; *Banerjee & Roychoudhury, 2017*; *Lakra et al., 2015*; *Liu, Wu & Wang, 2012*; *Zhang et al., 2015*, *2018*). The RT-qPCR results demonstrated that PgbZIPs showed substantial changes in gene expression levels upon exposure to low-temperature and MeJA stresses. PgbZIP gene expression may be significantly inhibited by low-temperature stress. Transgenic *A. thaliana* plants heterologously expressing *Triticum aestivum* bZIP6 (*TabZIP6*) and *Camellia sinensis* bZIP6 (*CsbZIP6*) upon exposure to low-temperature treatments showed reduced survival, increased relative conductivity, increased malondialdehyde content, and reduced soluble sugar content (*Cai et al., 2018*). This indicated that the overexpression of *TabZIP6* or *CsbZIP6* reduced the freezing tolerance of transgenic *A. thaliana* plants. Rice plants overexpressing *OsABF2* showed enhanced cold, salt, and drought tolerances (*Banerjee & Roychoudhury, 2017*). However, in the rice cold tolerance test, it was found that the *OsbZIP52* transgenic plant survival rate was 18%, and the survival rate of the wild type was 78.03%, which shows that *OsbZIP52* expression was negatively correlated with the low-temperature tolerance in rice (*Liu, Wu & Wang, 2012*). This is consistent with our findings that high PgbZIP expression levels may lead to a decrease in cold tolerance in *P. grandiflorus* plants so that a decrease in PgbZIP expression levels was observed after the low-temperature treatment. The main active constituents in *P. grandiflorus* are platycodins, and platycodin synthesis may be triggered by MeJA (*Huang et al., 2021*; *Kim et al., 2004*; *Scholz et al., 2009*). A significant increase in the final expression level of most

PgbZIPs occurred in *P. grandiflorus* after MeJA treatment. This suggests that PgbZIPs respond to MeJA induction, but whether PgbZIPs are involved in saponin content regulation in *P. grandiflorus* requires further study.

## ACKNOWLEDGEMENTS

At the point of finishing this article, I would like to express my sincere thanks to all those who have lent me a hand in the course of writing this article.

### Funding

This work was supported by the Key Natural Science Research Projects in Anhui Universities (2022AH050461, 2022AH040328), and the Project from Joint Research Center for Chinese Herbal Medicine of Anhui of IHM (yjzx2023001). The funders had no role in study design, data collection and analysis, decision to publish, or preparation of the manuscript.

### Grant Disclosures

The following grant information was disclosed by the authors:
Anhui Universities: 2022AH050461, 2022AH040328.
Chinese Herbal Medicine of Anhui of IHM: yjzx2023001.

### Competing Interests

The authors declare that they have no competing interests.

### Author Contributions

- Jizhou Fan conceived and designed the experiments, performed the experiments, analyzed the data, prepared figures and/or tables, authored or reviewed drafts of the article, and approved the final draft.
- Na Chen performed the experiments, analyzed the data, prepared figures and/or tables, and approved the final draft.
- Weiyi Rao conceived and designed the experiments, prepared figures and/or tables, and approved the final draft.
- Wanyue Ding analyzed the data, prepared figures and/or tables, authored or reviewed drafts of the article, and approved the final draft.
- Yuqing Wang analyzed the data, authored or reviewed drafts of the article, and approved the final draft.
- Yingying Duan analyzed the data, authored or reviewed drafts of the article, and approved the final draft.
- Jing Wu performed the experiments, authored or reviewed drafts of the article, and approved the final draft.
- Shihai Xing conceived and designed the experiments, prepared figures and/or tables, and approved the final draft.

## Data Availability

The genome database, the protein database, and related annotation files for *P. grandiflorus* are available at National Genomics Data Center: PRJCA003843.

The Hidden Markov Model (HMM) files for bZIP transcription factor conserved domains are available at Pfam: PF07716, PF00170, PF03131, and PF16326.

The raw data for RT-qPCR is available in the Supplemental File.

## Supplemental Information

Supplemental information for this article can be found online at http://dx.doi.org/10.7717/peerj.17371#supplemental-information.

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
