# Peer review of "Genome-wide analysis of bZIP transcription factors and their expression patterns in response to methyl jasmonate and low-temperature stresses in Platycodon grandiflorus"

_PeerJ, doi:10.7717/peerj.17371_

## Round 0.1 · original submission · Major Revisions

Thanks for submitting your work to PeerJ. Please improve the manuscript as per the reviewers' comments.

**Language Note:** The review process has identified that the English language must be improved. PeerJ can provide language editing services - please contact us at copyediting@peerj.com for pricing (be sure to provide your manuscript number and title). Alternatively, you should make your own arrangements to improve the language quality and provide details in your response letter. – PeerJ Staff

Reviewer 1 ·

Basic reporting

1. Please provide the full name once for the first time in the Abstract and Introduction, followed by abbreviations. For example, MeJA in line110 of the introduction does not appear for the first time, and the full name should be given in line91.
2. All the Latin names should use italics, such as Platycodon grandifloras in line135 and line141 does not have italics.
3.In Figure 7, the statistical results are lack of difference significance analysis. Please add these results.

Experimental design

1. Transcription factors directly or indirectly regulate gene expression to resist stress. Why choose low temperature and MeJA treatment? The author can make an in-depth analysis on this basis and elaborate on it according to the corresponding literature.
2.The greenhouse cultivation conditions for seedlings should provide detail information such as light duration, light intensity, humidity, etc.

Validity of the findings

no comment

Additional comments

no comment

Reviewer 2 ·

Basic reporting

Fan et al. conducted genome-wide identification of bZIP transcription factors in Platycodon grandiûorus. The features of these TFs were characterized. Expression patterns of ten bZIP TFs were analyzed in response to methyl jasmonate and low-temperature stresses. This study performed a comprehensive analyses of bZIP TFs in P. grandiûorus and provided potential candidate genes for better adaptation to abiotic stresses for P. grandiûorus cultivation. The structure of the manuscript is satisfied, however, english writing needs improvement. In addition, several issues need to be addressed before publication.

Experimental design

1) Please clarify the growing stage of seedlings used for stress treatment in line 126-128, such as three-leave stage or xx days after planting.

2) The experimental design for stress treatment in the Method section was not matching the results shown in Figure 7. The sampling time points were not matched.

3) The evolutionary relationship among bZIP TFs in P. grandiûorus was very different between the evolutionary tree in Figure1 and Figure5. Please clarify the discrepancy.

Validity of the findings

1) Line 237-239, the current findings are not necessarily draw the conclusion that 'gene duplication events are the main driving factors for expanding the PgbZIP gene family'. Gene duplication could be one of the driving factors for PgbZIP gene family expansion but more evidence is needed for concluding 'xxx are the main driving factors'.

2) Please add detailed genomic position information for each bZIP TF to supplementary Table S3.

3) Line 232, chromosomes other than Pgchr1, Pgchr2, Pgchr5, Pgchr6, and Pgchr7 also have many WGD events, please cautious on the conclusive sentence expression.

4) The diurnal or circadian rhythm effect could be discussed for the gene expression pattern in response to the two stresses. The immediate/early responsive or relatively late responsive genes could also be discussed in line 288-305.

Additional comments

1) Figure 3 shows the WGD events in the entire P. grandiûorus genome which is not directly related to the main topic of the paper. This figure can be combined with Figure 4 or moved to supplementary figures.

Reviewer 3 ·

Basic reporting

In the manuscript titled , “Genome-wide Analysis of bZIP Transcription Factors and Their Expression Patterns in Response to Methyl Jasmonate and Low-Temperature Stresses in Platycodon grandiflorus”, the authors have identified members of bZIP TF family in the named plant and performed some analyses thereafter. My comments are listed below:

The authors should provide the relevance of the study such as how identification of this particular transcription factor family members in this medicinal herb would be useful for anyone, if not just for the scientific community at least. Is there any link known that this gene family members may have an impact on any properties, or propagation or survival of this plant ? This should be described in the introduction.
In general, language is okay, however it needs proper phrasing of statements at a few instances, like Line 78, Line 79 (bZIP gene classification), Line 155, line 251, Line 255, Line 288 (How does stress increase genes ? ), Line 295, Line 302 etc…
Line 75: Some newer references needed.
In methodology: Can the author provide the model file for PF012498 ? Also, why only sequences from plant Arabidopsis were used for search ? Is P. grandiflorus related to A. thaliana ? If not, it would be useful to have a combined model from other organisms to search for gene family members in P. grandiflorus.
Further, it is important to verify the presence of the protein domain or at least presence of key amino acid residues in the gene family members. A multiple sequence alignment showing key amino acid residues that define the bZIP domain should be done and presented.
Has it been shown in the literature that 18s rRNA remains stable across the conditions that were used for RT-qPCR ? Provide a reference for the same. Else the data should be provided that shows the stability of this gene used as an internal control under the tested treatment conditions.
Line 155 unclear.
Line 200: Expand and explain GRAVY
Line 211: Provide in the text, the number of genes that do not cluster closely with Arabidopsis or form separate clades.
Line 223: Please use proper scientific molecular biology notations to denote parts of a chromosome.
Lines 238-241 : The analyses done is insufficient to conclude the evolutionary dynamics of this gene family based on single genes in one genome.
Line 246: Are there any names for the motifs? It is hard to figure out what motif is present / absent from the figure provided. With the bZIP domain in context, please explain what other domains are present and what could be their relevance ?
Line 262: Cannot understand what they mean by abscisic acid original or drought response original !
What was the criteria for choosing the bZIP genes for validation by RT-qPCR? Further, No data are shown on the number of replicates used for experiments and if any statistical tests are done. These are some absolute requirements for presenting results from such experiments.
Line 302, Which gene is being talked about in the text ?
The figure legends should detail the image and its description in a concise way. The figure legends are not at all proper.
In the discussion, except for multiple repetition of the phrase that the authors identified 75 gene family members , nothing else has been discussed. It's a mere repetition of the results. As said above one should describe what is the relevance and logic behind the study ? And, discuss the results in that context. How does the identification of this particular gene family members and their putative functions associate with the medicinal properties of the mentioned plant ? Or at least it’s tolerance to the growing conditions. At the moment it only appears completely artificial.

Experimental design

Please see the consolidated comments above.

Validity of the findings

Please see the consolidated comments above.

---

## Round 0.2 · Minor Revisions

Please address these changes and resubmit.
Thanks for submitting your work to PeerJ.

Reviewer 1 ·

Basic reporting

no comment

Experimental design

no comment

Validity of the findings

no comment

Additional comments

no comment

Reviewer 2 ·

Basic reporting

The overall quality was greatly improved in the revised version of manuscript. The structure, description of methods, delivery of results and English expression were enhanced.

Experimental design

The author basically addressed my question and make changes accordingly.

Validity of the findings

The author basically addressed my question and make changes accordingly.
One more comment:
Line 268-271. I suggest the author delete the last conclusive sentence in Line 271. Since the Ka/Ks was calculated within the PgbZIP gene family but not among closely related species, also only purifying selection detected, it's not right to draw conclusion that 'gene
271 duplication events could be one of the major driver of PgbZIP gene family expansion'.

Additional comments

none

---

## Round 0.3 · accepted · Accept

Congratulations. Thanks for submitting your work to PeerJ.
Your work has been Accepted for publication.

Reviewer 2 ·

Basic reporting

The revised manuscript addressed my concerns. I agree with its publication.

Experimental design

No further concerns

Validity of the findings

No further concerns

Additional comments

No further concerns